# Sodium Acetate Enhances Neutrophil Extracellular Trap Formation via Histone Acetylation Pathway in Neutrophil-like HL-60 Cells

**DOI:** 10.3390/ijms25168757

**Published:** 2024-08-11

**Authors:** Hiroyuki Yasuda, Yutaka Takishita, Akihiro Morita, Tomonari Tsutsumi, Naoya Nakagawa, Eisuke F. Sato

**Affiliations:** 1Department of Biochemistry, Faculty of Pharmaceutical Sciences, Suzuka University of Medical Science, 3500-3, Minamitamagaki, Suzuka 513-8670, Japan; yasuda20@mb.kyoto-phu.ac.jp (H.Y.); takisita.5239@gmail.com (Y.T.); morita-a@suzuka-u.ac.jp (A.M.); tutumi@suzuka-u.ac.jp (T.T.); n-naoya@suzuka-u.ac.jp (N.N.); 2Division of Pathological Sciences, Department of Pharmacology and Experimental Therapeutics, Kyoto Pharmaceutical University, Misasagi 5, Yamashina, Kyoto 607-8414, Japan

**Keywords:** neutrophil extracellular trap, NETosis, acetate, peptidyl arginine deiminase 4, histone acetylation, histone citrullination, immune response

## Abstract

Neutrophil extracellular trap formation has been identified as a new cell death mediator, termed NETosis, which is distinct from apoptosis and necrosis. NETs capture foreign substances, such as bacteria, by releasing DNA into the extracellular environment, and have been associated with inflammatory diseases and altered immune responses. Short-chain fatty acids, such as acetate, are produced by the gut microbiota and reportedly enhance innate immune responses; however, the underlying molecular mechanisms remain unclear. Here, we investigated the effects of sodium acetate, which has the highest SCFA concentration in the blood and gastrointestinal tract, on NETosis by focusing on the mechanisms associated with histone acetylation in neutrophil-like HL-60 cells. Sodium acetate enhanced NETosis, as shown by fluorescence staining with SYTOX green, and the effect was directly proportional to the treatment duration (16–24 h). Moreover, the addition of sodium acetate significantly enhanced the acetylation of Ace-H3, H3K9ace, and H3K14ace. Sodium acetate-induced histone acetylation rapidly decreased upon stimulation with the calcium ionophore A23187, whereas histone citrullination markedly increased. These results demonstrate that sodium acetate induces NETosis via histone acetylation in neutrophil-like HL-60 cells, providing new insights into the therapeutic effects based on the innate immunity-enhancing effect of dietary fiber.

## 1. Introduction

When activated, neutrophils undergo cell death via NETosis due to the release of extracellular traps (NETs). This form of cell death differs from apoptosis and necrosis in terms of changes in cell morphology and mechanism [1,2]. Several studies have reported that increased NETosis induces inflammation, thrombosis, or autoimmune diseases [3,4,5]. We have previously reported that estradiol activates NETosis [6], which may be associated with the predominance of diseases such as systemic lupus erythematosus and rheumatoid arthritis in females. Recently, it has been reported that excessive induction of NETosis causes inflammation and thrombosis during SARS-CoV-2 infection [7,8,9,10]. Therefore, understanding and suppressing NETosis will play an important role in the development of therapeutic agents against various diseases.

There are two types of NETosis: NADPH oxidase (NOX)-dependent and NOX-independent [11,12]. NOX-dependent NETosis is induced by phorbol myristate acetate (PMA), LPS, or bacteria [13], whereas NOX-independent NETosis is induced by the calcium ionophore A23187 or ionomycin in the presence of calcium [14]. As reactive oxygen species (ROS) production and histone citrullination are important factors in NETosis, the development of treatments for excessive NET formation relies on antioxidants and the peptidyl arginine deiminase 4 (PAD4) inhibitor, Cl-amidine. Moreover, these treatments are successful in suppressing NETosis [15,16,17]. NETosis is essentially a biological defense mechanism and its activation is beneficial for the innate immune response.

Recently, it has become clear that short-chain fatty acids (SCFAs) produced by the gut microbiota play a crucial role in host health. Among SCFAs, acetic acid ingestion has been shown to help prevent lifestyle-related diseases by regulating acetic acid metabolism and signal transduction via receptors. SCFAs bind directly to GPR43 [18]. Moreover, SCFAs promote insulin-mediated fat accumulation [18], inhibit histone deacetylase activity [19], and regulate T-cell differentiation in adipose tissues via GPR43 [20]. In addition, SCFAs affect innate immune cells, inducing inflammatory mediator production by macrophages and also ROS production and phagocytosis by neutrophils [21,22,23]. Although the effects of SCFAs on various processes have been studied, their effects on NET function are unknown. This study aimed to identify the acetylated histone that contributes to the regulation of extracellular traps in neutrophils.

Acetic acid has previously been reported to promote the extracellular trapping of neutrophils via signaling from GPR43 receptors present on the cell membrane [24,25]. However, acetate has also been reported to affect cellular metabolism not only through the GPR43-mediated pathway [24,25] but also by being taken up intracellularly by monocarboxylic acid transporters [26]. Acetic acid taken up by cells is converted to acetyl CoA by acetyl CoA synthase 2 and to histone acetylation [27]. Extracellular neutrophil trapping has also been reported to be affected by histone acetylation [28,29]. However, it is unclear whether the effects of acetic acid on cells, particularly on extracellular traps, occur via the GPR43-mediated pathway or histone acetylation. Therefore, in this study, we analyzed whether the effect of acetic acid on extracellular traps was affected by histone acetylation.

We used neutrophil-like HL-60 cells (nHL-60 cells) differentiated in dimethyl sulfoxide (DMSO) to investigate the association between sodium acetate and calcium ionophore-induced NOX-independent NETosis in vitro. This is the first study to explore the mechanisms underlying the relationship between sodium acetate and NOX-independent NETosis in differentiated neutrophils.

## 2. Results

### 2.1. Sodium Acetate Treatment Increased NOX-Independent NETosis in nHL-60 Cells

We investigated whether NETosis increased in nHL-60 cells concomitantly treated with 1.25% DMSO and sodium acetate. Our data indicated a significant increase in NETosis following sodium acetate (10 mM) treatment in nHL-60 cells compared to untreated nHL-60 cells (Figure 1a,b). The effect of sodium acetate on NETosis was also directly proportional to treatment duration (16–24 h).

### 2.2. Sodium Acetate Treatment Did Not Increase ROS Production in nHL-60 Cells

To elucidate the role of sodium acetate in NETosis, we analyzed the production of ROS, which is an NOX-dependent NETosis indicator. Following A23187 treatment for 1 h, ROS levels were unchanged in the treated nHL-60 cells compared to those in the untreated control cells (Figure 2). These data indicate that the increase in NETosis in nHL-60 cells treated with sodium acetate occurred specifically via NOX-independent pathways.

### 2.3. Effect of ACSS2 (Acetyl-CoA Synthetase 2) Inhibitor on Extracellular DNA Release

We analyzed whether sodium acetate uptake affected neutrophil extracellular traps using an ACSS2 inhibitor (acetyl-CoA synthetase 2). As shown in Figure 3, the ACSS2 inhibitor (Selleck, Houston, TX, USA) suppressed the sodium acetate-induced increase in extracellular DNA. Therefore, the increase in neutrophil extracellular traps caused by sodium acetate is likely due to sodium acetate uptake into cells.

### 2.4. Expression of ACSS2, MCT-1, and MCT-4

Neutrophils express monocarbonic acid transporter (MCT) [26,30], an acetate transporter. Therefore, we examined whether similar transporters are expressed in differentiated neutrophil-like HL-60 cells. As shown in Figure 4, neutrophil-like HL-60 cells expressed MCT1 and MCT4, but their expression was unchanged upon the addition of sodium acetate. Furthermore, for sodium acetate to be taken up by cells for histone acetylation, it must be converted to acetyl-CoA. Therefore, we analyzed the acetyl-CoA synthase 2 (ACSS2) expression in differentiated neutrophil-like HL-60 cells. As shown in Figure 4, neutrophil-like differentiated HL-60 cells expressed ACSS2, which was unchanged upon adding sodium acetate. These results suggest that neutrophil-like differentiated HL-60 cells can take up extracellular acetate and produce acetyl-CoA, which serves as a substrate for histone acetylation.

### 2.5. Sodium Acetate Treatment Increased Histone Acetylation

As shown in Figure 5, treatment with sodium acetate significantly increased histone acetylation in nHL60 cells. Histone acetylation did not occur much after 1 h but was noticeable after 24 h of culture. This effect became more pronounced as the sodium acetate concentration increased from 1 mM to 10 mM. When histone acetylation was investigated at the N-terminus of H3, significant acetylation was observed at H3K9 and H3K14, as shown in Figure 6.

### 2.6. Histone Acetylation Decreased after Stimulation of A23187

Previous studies using histone deacetylase inhibitors have shown that histone acetylation causes nuclear decondensation and induces neutrophil extracellular traps. Therefore, the increase in histone acetylation caused by sodium acetate treatment may directly lead to decondensation. We then analyzed the histone acetylation state after the stimulation of nHL-60 cells, in which histone acetylation increased with sodium acetate treatment. Surprisingly, as shown in Figure 7, histone acetylation instantly disappeared when stimulated with A23187, although it increased upon sodium acetate treatment.

### 2.7. Sodium Acetate Treatment Increased NETosis by Increasing Histone Citrullination

To analyze the mechanism underlying the increase in neutrophil extracellular traps due to sodium acetate treatment, we investigated histone citrullination and the expression of its enzyme PAD4, which is directly involved in chromatin decondensation. As shown in Figure 8a, A23187 treatment-induced histone citrullination was markedly enhanced by sodium acetate treatment. However, treatment with sodium acetate did not affect PAD4 expression (Figure 8b). These results suggest that sodium acetate enhanced NETosis via histone citrullination. These results suggest that the acetylation of specific lysine residues in histones may directly or indirectly accelerate histone citrullination by PAD4.

## 3. Discussion

We found that sodium acetate alone did not induce NETosis; however, long-term treatment with 10 mM sodium acetate combined with A23187 enhanced NETosis induction. These treatments did not affect the expression of PAD4 but enhanced histone acetylation and citrullination. Surprisingly, histone acetylation instantly disappeared when stimulated with A23187, although it increased upon sodium acetate treatment (Figure 6). Therefore, the increase in NETosis could be attributed to an increase in histone citrullination. 

In this study, sodium acetate did not induce NETosis but increased NOX-independent NETosis. Sodium acetate induces NETosis in isolated human peripheral blood neutrophils when cultured at a physiological concentration of 100 μM [25], but such results were not obtained in this study. As this study used DMSO-differentiated HL-60 cells, the data may differ from those of isolated human peripheral blood neutrophils [24,25]. Additionally, NETosis has not yet been reported in human blood, even after increasing the concentration of acetic acid in the blood of healthy participants. However, in a previous study, the reaction solution did not contain serum, and acetic acid alone induced NETosis [24]. In contrast, the reaction solution used in our study was RPMI1640 culture solution containing 10% FCS, which closely simulates human physiological conditions. Ohbuchi et al. showed that acetic acid reduces PMA-induced NETosis in isolated human blood neutrophils [25]. 

Recently, it was reported that SCFAs, including acetic acid, enhance innate immunity in various ways, such as by activating inflammasomes in macrophages [31]. This study showed that SCFAs activate inflammasomes even in GPR43 knockout mice, demonstrating that the reaction is not mediated by GPR43. It is transported into cells by a monocarboxylic acid transporter. Because the monocarboxylic acid transporter is present in neutrophils, it is possible that intracellular inflammasomes are activated and NETosis is promoted without GPR43 involvement. Moreover, it has been reported that the activation of the neutrophil inflammasome enhances NETosis [32,33,34]. 

Sodium acetate treatment increased histone acetylation in a concentration-dependent manner (Figure 5). Thomas and Denu [35] reported that histone acetylases are activated when acetic acid is taken up by the cells. Furthermore, it has been reported that histone deacetylases are inhibited by acetic acid administration [36]. Thus, acetic acid administration may sufficiently increase histone acetylation of intracellular chromatin. Previous studies [28,29] have reported that stimulant-dependent neutrophil extracellular trapping is increased by increasing histone acetylation through drug treatment, which inhibits histone deacetylases. However, the acetylation status of histones following stimulation has not yet been reported. In these studies, decondensation by acetylation was thought to increase extracellular trap formation. However, in this study, it was found that even though histone acetylation is increased by acetic acid administration, acetylation is markedly decreased in a short time after stimulation with A23187 (Figure 7). Stimulation with A23187 causes an influx of calcium ions into the cell. This induces the activation of PAD4, an enzyme that causes histone citrullination by inducing an influx of calcium ions into the cell. Previous studies [37] have reported that PAD4 forms a complex with histone deacetylase HDAC2. Therefore, the activation and binding of PAD4 to histones simultaneously induce the binding of histone deacetylases to histones. Saiki et al. [38] reported that the binding of PAD4 to the N-terminus of histone H3 was significantly increased by acetylation in in vitro experiments using peptides. Thus, it is possible that histone H3 acetylation in this study increased the binding of PAD4 and citrullination, and that HDAC2, which forms a complex with PAD4, increased deacetylation. As shown in Figure 6, the acetylation of histone H3 by sodium acetate treatment increased at H3K9 and H3K14. The acetylation site responsible for the increased binding of PAD4 requires further analysis.

It has been reported that sodium propionate and butyrate salts are present in the gastrointestinal tract to some extent, but only acetate is usually found in the blood at an effective concentration of approximately 0.5 mM [24]. However, since the serum acetate concentration varies with diet, histone acetylation-mediated enhancement of NETosis may only occur when the acetic acid concentration crosses the minimum threshold.

The effect of 10 mM sodium acetate used in this experiment was expected to be similar to that observed in in vivo experiments. As the main sources of sodium acetate are food intake and metabolism by the gut microbiota [39], the bloodstream around the gastrointestinal tract may contain sodium acetate at concentrations close to 10 mM [24]. Furthermore, the liver produces acetate under normal physiological conditions as well as large amounts of acetic acid under certain conditions, such as high alcohol consumption [40], starvation, or diabetes [41]. Moreover, certain bacteria, such as *Staphylococcus aureus*, produce acetate during substrate-level phosphorylation [23]. Thus, in the event of a bacterial infection, the accumulated neutrophils will respond to high localized concentrations, possibly activating NETosis.

When NETosis increases, various diseases are exacerbated. Therefore, enhancing NETosis with sodium acetate may aggravate the pathological conditions. However, in many cases, these immunostimulatory events alleviate subsequent diseases. Schlatterer et al. reported that enhanced neutrophil chemotaxis, ROS generation, cytokine production, and phagocytosis via acetic acid alleviated the pathology of sepsis caused by bacterial infection in mice [23]. Therefore, increasing the acetic acid concentration in the blood through dietary intake is advantageous for the human body owing to its immunostimulatory effect and the enhancement of NETosis and innate immunity.

In conclusion, the results of this study indicate that A23187-induced NETosis in nHL-60 cells was enhanced through treatment with sodium acetate via histone acetylation during histone citrullination. Additionally, the regulation of sodium acetate-induced NETosis was independent of NOX-derived ROS production. This study provides a foundation for future studies investigating the significance of NET formation in the innate immune response.

## 4. Materials and Methods

### 4.1. Cell Culture

HL-60, a human promyelocytic leukemia cell line (RCB3683; RIKEN BioResource Center, Ibaraki, Japan), was cultured in an RPMI 1640 medium (Nacalai, Kyoto, Japan) containing 10% (*v*/*v*) inactivated fetal bovine serum and 1% penicillin/streptomycin (Fujifium-Wako, Osaka, Japan) [42]. The cells were maintained at 37 °C in a humidified incubator (5% CO_2_), and the culture medium was replaced every 2 d. To differentiate HL-60 cells into neutrophil-like cells, the cells were cultured with 1.25% DMSO for 3 d [43].

### 4.2. Preparation of Sodium Acetate

A 200 mM stock solution of acetate was prepared using sodium acetate (Fujifilm-Wako, Osaka, Japan) in phosphate-buffered saline (PBS), and the pH was adjusted to 7.4 using NaOH. This neutral acetate solution was then filtered using a 0.22 μm syringe filter.

### 4.3. SYTOX Green NETosis Assay

NETosis was analyzed using SYTOX green fluorophotometry (Invitrogen, Tokyo, Japan) [14]. The nHL-60 cells, either untreated or treated with 10 mM sodium acetate for 24 h, were seeded with SYTOX green, a cell-impermeable nucleic acid dye, in 96-well plates at a density of 5 × 10^4^ cells/well. The plates were divided into four different groups: one group was treated with 10 mM sodium acetate, another was treated with 10 μM A23187 (Fujifilm-Wako, Osaka, Japan), the third was treated with 10 μM A23187 and 10 mM sodium acetate, and the fourth was the control group (n = 6 per group) treated with 10 mM NaCl for comparison with the osmolarity changes found in treatment with 10 mM sodium acetate. After adding 10 μM of A23187 to two of the groups, changes in green fluorescence in all of the groups were measured every 1 h using SpectraMax^®^ (485 nm excitation, 525 nm emission; Molecular Devices Japan, Tokyo, Japan). To determine the total DNA concentration, the nHL-60 cells were lysed with 1% (*v*/*v*) Triton X-100 (Fujifilm-Wako, Osaka, Japan) and changes in fluorescence were recorded. All values were standardized using the total DNA concentration in each experiment.

### 4.4. NET Visualization

To induce NET formation, the nHL-60 cells, untreated or treated with 10 mM sodium acetate for 24 h, were seeded at 2 × 10^4^ cells in flexiPERM^®^ (pore size: 1.8 cm^2^) (Sarstedt, Tokyo, Japan) on slide grass and then incubated with 10 μM A23187. SYTOX green (5 μM) was then added, and the cells were further incubated for 5 min. Changes in fluorescence were observed using a confocal microscope (OLYMPAS, Tokyo, Japan). 

### 4.5. ROS Measurement

The nHL-60 cells, untreated or treated with 10 mM sodium acetate for 24 h, were seeded at 4 × 10^3^ cells per 200 μL of PBS and incubated at 37 °C with 500 μM L-012 (Fujifilm-Wako, Osaka, Japan). In groups with or without 10 μM A23187, chemiluminescence was measured post-treatment using a Microplate Luminometer (Orion II; Roche Diagnostics K.K., Tokyo, Japan).

### 4.6. Quantification of Extracellular DNA

Briefly, the nHL-60 cells treated with or without sodium acetate for 24 h were seeded at a density of 1 × 10^6^ cells/mL in 96-well plates. After the addition of 10 μM A23187 for 3 h, the cells were treated with 20 U/mL Micrococcal Nuclease (MNase, New England Biolabs Japan, Tokyo, Japan) for 20 min at 37 °C. The supernatants including DNA were collected after centrifugation at 200× *g* for 8 min at 4 °C. The extracellular DNA was quantified using SYTOX green.

### 4.7. Total Cell Protein Extraction

Total cell protein was extracted as described previously [12]. Briefly, the cells were gently suspended in a lysis buffer (20 mM HEPES-NaOH, pH 7.8, containing 15 mM KCl, 2 mM MgCl_2_, 0.5% NP-40, and a protease inhibitor cocktail). After centrifugation at 15,000× *g* for 20 min, the supernatant was collected and protein concentrations were estimated using a BCA protein assay.

### 4.8. Histone Extraction

Histones were extracted as previously described [44]. First, the cells were lysed in an extraction buffer (containing 0.1 M tris-HCl, pH 7.5, 0.15 M NaCl, 1.5 mM MgCl_2_, 0.65% NP-40, and a protease inhibitor cocktail (Nakarai-Tesque, Kyoto, Japan)). After centrifugation at 13,200× *g* for 10 s, the pellets obtained were mixed with 0.2 M H_2_SO_4_ and further centrifuged at 13,200× *g* for 20 min. The resulting supernatant was mixed with 100% trichloroacetic acid and centrifuged at 13,200× *g* for 20 min. The pellets were washed with acetone, centrifuged again at 13,200× *g* for 5 min, and then dissolved in 0.45 M tris-HCl (pH 8.8) containing 2% SDS, 6% 2-mercaptoethanol, and 0.01% bromophenol blue.

### 4.9. Western Blotting

Nuclear extracts (20 μg) were mixed with a sample buffer solution (containing 62.5 mM tris-HCl, pH 6.8, 2% SDS, 5% glycerol, 0.8% 2-mercaptoethanol, and 0.012% bromophenol blue) and native histone extracts. The resultant mixture was boiled for 5 min and then separated via SDS-PAGE on 12.5%w/v polyacrylamide gels. The separated proteins were electrophoretically transferred to PVDF and nitrocellulose membranes. Nonspecific binding was blocked using a solution of 5% skim milk in TBST (containing 20 mM tris-HCl, pH 7.5, 0.15 M NaCl, and 0.05% Tween 20). The membranes were then incubated with primary antibodies against ACSS2 (Proteintech (Rosemont, IL, USA), cat 16087-1-AP, 1:1000), MCT-1 (Proteintech, cat 20139-1-AP, 1:1000), MCT-4 (Proteintech, cat 22787-1-AP, 1:1000), acetylated lysine (Cell Signaling Technology (Danvers, MA, USA), cat #9441, 1:1000), Acetyl-Histone H3 (Lys9) (Cell Signaling Technology, cat #9649, 1:1000), Acetyl-Histone H3 (Lys14) (Cell Signaling Technology, cat #7627, 1:1000), Acetyl-Histone H3 (Lys18) (Cell Signaling Technology, cat #13998, 1:1000), Acetyl-Histone H3 (Lys27) (Cell Signaling Technology, cat #8173, 1:1000), histone H3 (citrulline R2+R8+R17) (Abcam (Cambridge, UK), cat ab5103, 1:1000), and PAD4 (Abcam, cat ab50332, 1:1000). After washing with TBST, the membranes were incubated with anti-rabbit or anti-mouse IgG-peroxidase antibodies (1:3000). The protein bands were detected using ImmunoStar^®^ Zeta (Wako Pure Chemical Industries, Ltd., Osaka, Japan) and visualized using Amersham Imager 600 (GE Healthcare Life Sciences, Tokyo, Japan).

### 4.10. Statistical Analysis

The data from the experiments are presented as the mean ± SD. Statistical analysis was performed using Student’s t-test or two-way analysis of variance (ANOVA) with Tukey–Kramer post hoc comparisons. Statistical significance was set at *p* < 0.05.

## Figures and Tables

**Figure 1 ijms-25-08757-f001:**
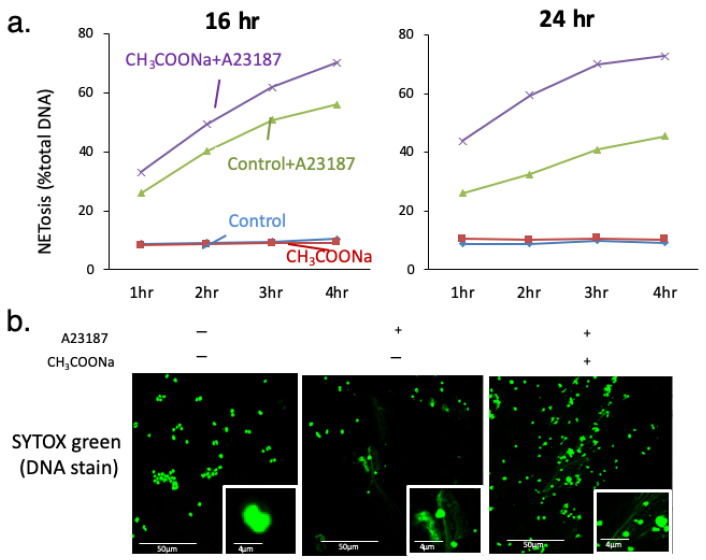
Sodium acetate increases A23187-induced neutrophil extracellular trap formation (NETosis) (NOX-independent NETosis). (**a**) After treatment with or without 10 mM sodium acetate during differentiation, NETosis levels in nHL-60 cells treated with (+) or without (−) 10 μM A23187 for 4 h were analyzed through SYTOX green assay (*n* = 5). Control was not treated with sodium acetate. 16 hr and 24 hr: incubation time of sodium acetate. After sodium acetate incubation, cells were stimulated by A23187. (**b**) After treatment with or without 10 mM sodium acetate during differentiation, NETosis images were obtained using confocal microscopy of SYTOX green (DNA)-stained nHL-60 cells treated with (+) or without (−) 10 μM A23187 for 2 h.

**Figure 2 ijms-25-08757-f002:**
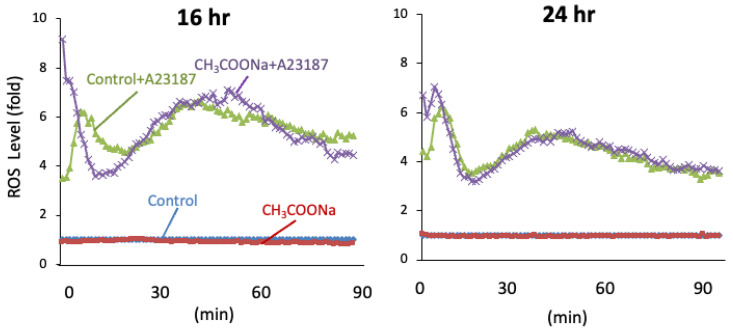
Sodium acetate treatment did not increase ROS production in nHL-60 cells after treatment with or without 10 mM sodium acetate during differentiation. Control was not treated with sodium acetate. 16 hr and 24 hr: incubation time of sodium acetate. After sodium acetate incubation, cells were stimulated by A23187. Quantitative chemiluminescence analysis of ROS production in nHL-60 cell culture incubated with or without 10 μM A23187 for 90 min was performed. Data are shown as mean ± SD (*n* = 3).

**Figure 3 ijms-25-08757-f003:**
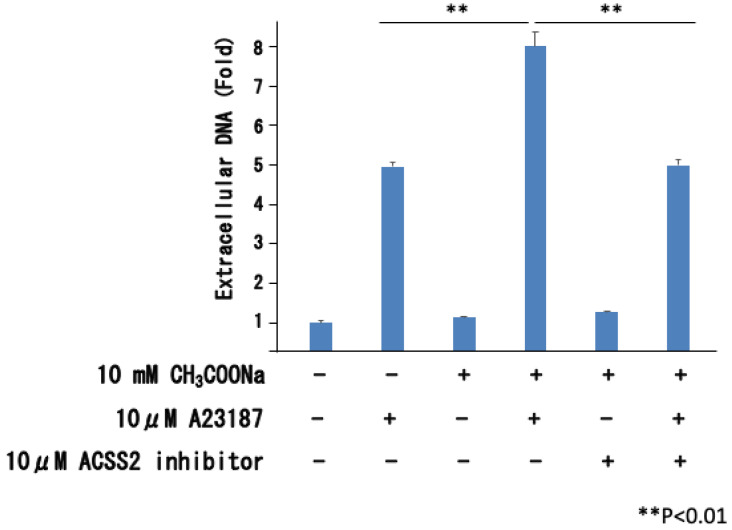
Effect of ACSS2 inhibitor on release of extracellular DNA. After treating nHL-60 cells with or without 10 μ M ACSS2 inhibitor during sodium acetate treatment, extracellular DNA was isolated and measured by performing extracellular DNA quantification assay. +: Addition of reagents, −: No treatment. Data are shown as mean ± SD (*n* = 3).

**Figure 4 ijms-25-08757-f004:**
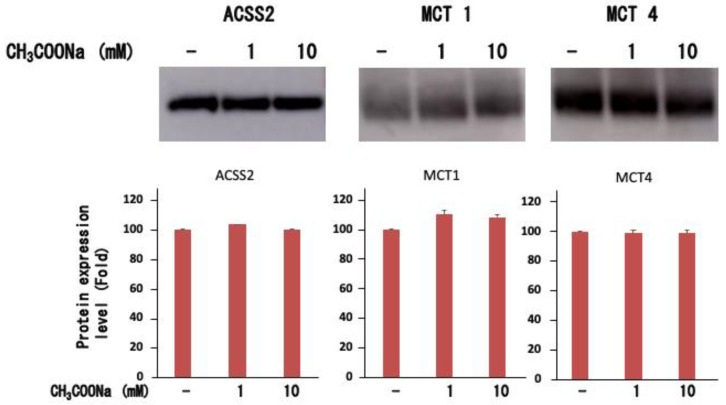
Expression of ACSS2, MCT-1, and MCT-4 in nHL-60 cells. After treatment with or without sodium acetate (1 or 10 mM) for 24 h, ACSS2, MCT-1, and MCT-4 expression levels in nHL-60 cells were analyzed using Western blotting. −: No treatment. Data are shown as mean ± SD (*n* = 3).

**Figure 5 ijms-25-08757-f005:**
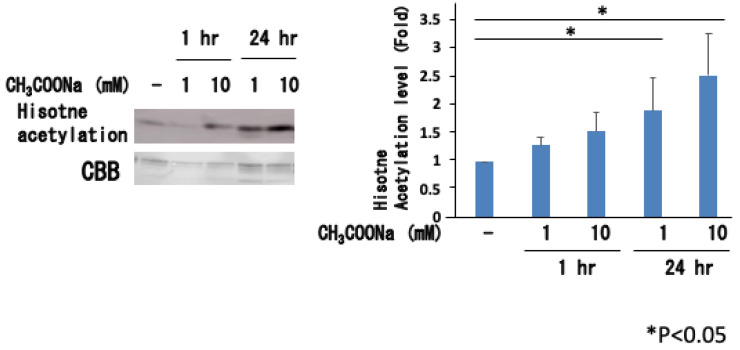
Sodium acetate enhanced histone acetylation. The nHL-60 cells were incubated with or without sodium acetate for 1 or 24 h. −: The control was not treated with sodium acetate. Following histone extraction, histone acetylation levels were analyzed using Western blotting. Loading of the histones was monitored using Coomassie staining (denoted as CBB). The data are shown as the mean ± SD (*n* = 3).

**Figure 6 ijms-25-08757-f006:**
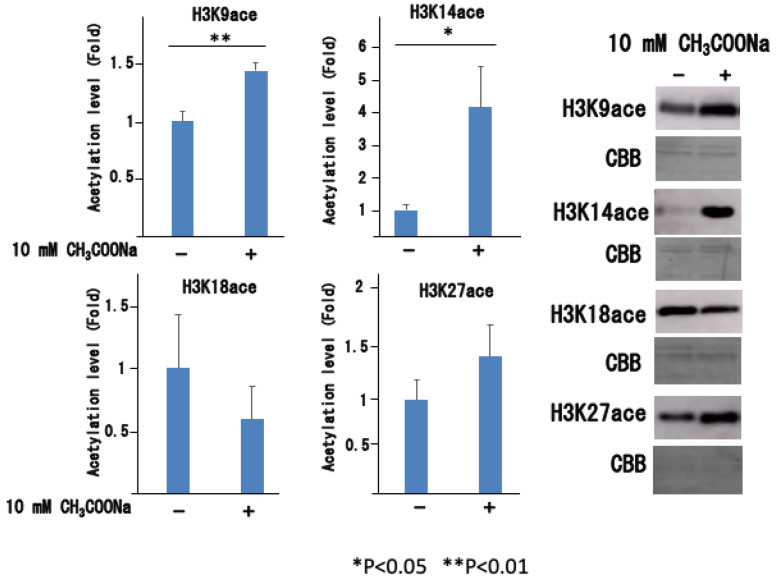
Histone H3 acetylation in nHL-60 cells. The nHL-60 cells were incubated with or without sodium acetate for 24 h. Following histone extraction, histone acetylation (denoted as H3K9ace, H3K14ace, H3K18ace, and H3K27ace) levels were analyzed by Western blotting using specific antibodies. −: The control was not treated with sodium acetate. +: Addition of 10 mM CH_3_COONa. Loading of the histones was monitored using Coomassie staining (denoted as CBB). The data are shown as the mean ± SD (*n* = 3).

**Figure 7 ijms-25-08757-f007:**
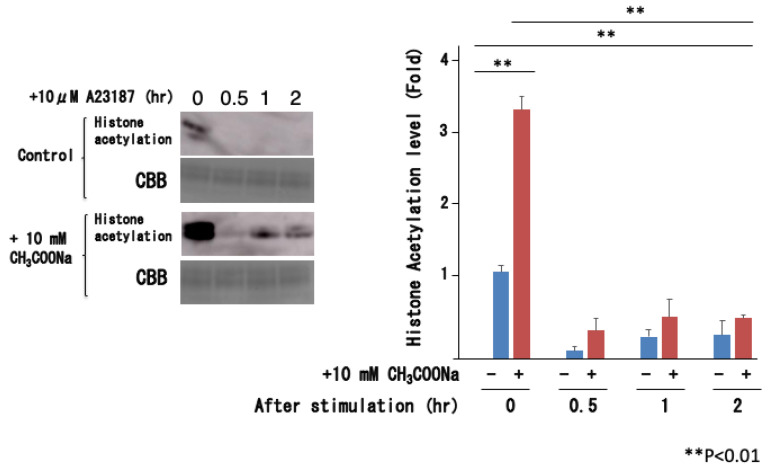
Histone acetylation levels before and after A23187 stimulation. The nHL-60 cells were incubated with or without 10 mM sodium acetate for 24 h. Then, the cells were stimulated with 10 μM A23187. Following histone extraction, histone acetylation levels were analyzed by Western blotting. The control was not treated with sodium acetate. +: Addition of 10 mM CH_3_COONa, −: No treatment. Loading of the histones was monitored using Coomassie staining (denoted as CBB). The data are shown as the mean ± SD (*n* = 3).

**Figure 8 ijms-25-08757-f008:**
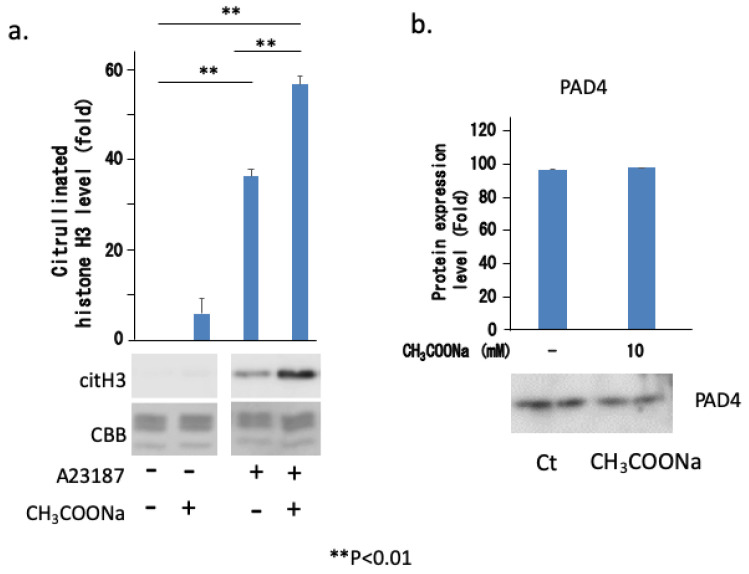
Sodium acetate enhanced histone citrullination but not PAD4 expression. (**a**) nHL-60 cells (treated with or without 10 mM sodium acetate) were treated with or without 10 μM A23187 for 3 h. Following histone extraction, citrullinated histone H3 (denoted as citH3) protein levels were analyzed via Western blotting. Loading of histones was monitored using Coomassie staining (denoted as CBB). +: Addition of reagents, −: No treatment. (**b**) nHL-60 cells were incubated with or without 10 mM sodium acetate. After cell extraction, PAD4 expression was determined through Western blotting. Ct: Control was not treated with sodium acetate. Data are shown as mean ± SD (*n* = 3).

## Data Availability

The data supporting the findings of this study are available upon request from the corresponding author.

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
