# Peer review of "Sodium Acetate Enhances Neutrophil Extracellular Trap Formation via Histone Acetylation Pathway in Neutrophil-like HL-60 Cells"

_ijms, 2024, doi:10.3390/ijms25168757_

Round 1

Reviewer 1 Report

Comments and Suggestions for Authors

Dear Authors,

Your paper is well written and understandable; however it rises some concerns that need to be addressed:

Minor issues: 

-Line 109 twice “..” near extracelular DNA

-Line 223 there is a double space between A23187 and Figure7

Major issues:

-Figure 1- a and b, it is not clear the number of independent experiments, needs to be known.

- b) There is a need to complete the figure and add the scale

-Figure 4 lack of house keeping protein

- The data mentioned at lines 202 and 237 needs to be shown. It may validate the results from the paper. 

- figure 8, it is hard to understand the third bar of the graph and its comparison with the second bar based on the image of the gel shown. It looks like the samples were not run in the same gel and so cannot be compared. Moreover, there are no indication of raw images from the gel of figure 8 in the original images file.

Author Response

Comment 1: -Line 109 twice “..” near extracelular DNA

Response 1:  We have corrected it accordingly.

Comment 2:-Line 223 there is a double space between A23187 and Figure7

Response 2:  We have corrected it accordingly.

Major issues:

Comment 3: -Figure 1- a and b, it is not clear the number of independent experiments, needs to be known.

Response 3:  We have corrected it accordingly.  The number of independent experiments was 5. We added it in the figure 1.

Comment 4:- b) There is a need to complete the figure and add the scale

Response 4:  We have corrected it accordingly.

Comment 5:-Figure 4 lack of house keeping protein

Response 5: We normalize Western blotting by equalizing the total protein amount of the electrophoresed samples. Therefore, in Figure 4, the comparisons are made by maintaining a constant total protein amount for the electrophoresed samples. When electrophoresis was performed with equal amounts of total protein, as shown in the supplementary figure, the expression levels of the housekeeping protein β-actin did not change with or without the addition of sodium acetate.

Comment 6:- The data mentioned at lines 202 and 237 needs to be shown. It may validate the results from the paper. 

Response 6: Revised manuscript line 220: While these findings might provide effect of sodium acetate on NOX-independent netosis pathway, our study aim was to analyze the effect of A23187-induced netosis but not to PMA-induced NOX-dependent pathway. It is known that DMSO-differentiated nHL-60 cells are poorly responsive to PMA, and in our laboratory, they are also poorly responsive to PMA, making analysis difficult.  Thus, we could not present in this paper.  Moreover, it takes time to establish the data. However, if the editor and the reviewer prefer, we are happy to include these data in the manuscript.

Revised manuscript line 255: While these findings might provide effect of sodium acetate on neutrophil extracellular traps of nHL-60 cells, our study aim was to analyze the effect of sodium acetate but not to other short chain fatty acids. The concerning data is too preliminary, so we could not present in this paper. And it takes time to establish the data. However, if the editor and the reviewer prefer, we are happy to include these data in the manuscript.

Comment 7:- figure 8, it is hard to understand the third bar of the graph and its comparison with the second bar based on the image of the gel shown. It looks like the samples were not run in the same gel and so cannot be compared. Moreover, there are no indication of raw images from the gel of figure 8 in the original images file.

Response 7: The positions of the bands used in the figure are indicated on the original image files.

Reviewer 2 Report

Comments and Suggestions for Authors

General comments:

It is a well-structured study with comprehensible results. The introduction contains the necessary information for the main part of the present study. The methodological approach is good and well thought. The discussion is well structured and comprehensible. The revision of some points would make the results easier to understand and enhance the publication.

Specific comments:

In some of the references the numbers are missing: Line 371, 381, 401.

Page 2, line 46: “NOX-dependent NETosis is induced by phorbol myristate acetate (PMA), whereas NOX-independent NETosis is induced by the calcium ionophore A23187 in the presence of calcium” That sound like NOX-dependent NETosis is induced only by PMA and NOX-independent only by A23187. That is not the case, there are further inducer like LPS and bacteria of the NOX-dependent NETosis and e.g. Ionomycin induce the NOX-independent one. Please complement this within the text with the corresponding references.

Figure 1A shows the NETose formation over a period of 1 to 4 hours. Within the figure caption you write “treated with (+) or without (-) 10 μM A23187 for 4 h were analyzed”. What do the 16 and 24 hours mean? That point is not clear. Please make it understandable for the reader without having to read the methods. The way you do it in figure 7 is nice and clear. You need to add scale bars to the fluorescence images.

Figure 2 shows ROS measurement over period of 1h and is also described in the text. What do the 16 and 24 hours above the graph mean? That point is not clear. Please make it understandable for the reader without having to read the methods. The way you do it in figure 7 is nice and clear.

Introduce the ACSS2 abbreviation already on Page 3, line 108, using it the first time.

Page 3, line 109: Delete the second dot. “in extracellular DNA..”

Page 4, line 118. Please provide a reference for the information “Neutrophils express monocarbonic acid transporter (MCT), an acetate transporter.”

Point 2.5 and Figure 5: When was the histone acetylation measured of the control, which was not treated with sodium acetate? After 1h or 24h? That should be added.

Pge 5, line 139. In the text you write, that “significant acetylation was observed at H3K9, H3K14, and H3K27” but in figure 6 you indicate significance only for H3K9, H3K14. Please clarify that. In the shown full Western blot of Figure 5 H3K27, the 1. experiment shows opposite results to the 2. experiment. In the 3. experiment the bands are looking very similar. I doubt an acetylation increase of H3K27 with CH3COONa. Please recheck this.

Figure 8: It would be easier to read and understand the treatments, if you label the treatment of A23187 an CH3COONa as you did it in figure 1b for the images.

Specific comments to Supplemented Figures:

I think Figure 3 should actually be named Figure 4. Is that correct? If so, please change it.

Please check all the Figure numbers, there are some more mistakes.

Please indicate the size assignment of the protein marker.

Author Response

Comment 1: In some of the references the numbers are missing: Line 371, 381, 401. 

Response 1:  We have corrected it accordingly.

Comment 2: Page 2, line 46: “NOX-dependent NETosis is induced by phorbol myristate acetate (PMA), whereas NOX-independent NETosis is induced by the calcium ionophore A23187 in the presence of calcium” That sound like NOX-dependent NETosis is induced only by PMA and NOX-independent only by A23187. That is not the case, there are further inducer like LPS and bacteria of the NOX-dependent NETosis and e.g. Ionomycin induce the NOX-independent one. [1]Please complement this within the text with the corresponding references. 

Response 2 :  We have corrected it accordingly.

Line 46, 47: There are two types of NETosis: NADPH oxidase (NOX)-dependent and NOX-independent [13]. NOX-dependent NETosis is induced by phorbol myristate acetate (PMA), LPS, or bacterias, whereas NOX-independent NETosis is induced by the calcium ionophore A23187 or ionomycin in the presence of calcium [14].

Comment 3: Figure 1A shows the NETose formation over a period of 1 to 4 hours. Within the figure caption you write “treated with (+) or without (-) 10 μM A23187 for 4 h were analyzed”. What do the 16 and 24 hours mean? That point is not clear. Please make it understandable for the reader without having to read the methods. The way you do it in figure 7 is nice and clear. You need to add scale bars to the fluorescence images. 

Response 3:  We have corrected it accordingly.

Line 90: Figure 1. Sodium acetate increases A23187-induced neutrophil extracellular trap formation (NETosis) (NOX-independent NETosis). (a) After treatment with or without 10 mM sodium acetate during differentiation, NETosis levels in nHL-60 cells treated with (+) or without (-) 10 μM A23187 for 4 h were analyzed through a SYTOX green assay (n=6). 16hr, 24hr: incubation time of sodium acetate. After sodium acetate incubation , the cell was stimulated by A23187 (b) After treatment with or without 10 mM sodium acetate during differentiation, NETosis images were obtained using confocal microscopy of SYTOX green (DNA) stained nHL-60 cells treated with (+) or without (-) 10 μM A23187 for 2 h.

Comment 4: Figure 2 shows ROS measurement over period of 1h and is also described in the text. What do the 16 and 24 hours above the graph mean? That point is not clear. Please make it understandable for the reader without having to read the methods. The way you do it in figure 7 is nice and clear.

Response 4:  We have corrected it accordingly.

Line 108: Figure 2. Sodium acetate treatment did not increase ROS production in nHL-60 cells. After treatment with or without 10 mM sodium acetate during differentiation. 16hr, 24hr: incubation time of sodium acetate. After sodium acetate incubation, the cell was stimulated by A23187. Quantitative chemiluminescence analysis of ROS production in the nHL-60 cell culture incubated with or without 10 μM A23187 for 90 min was performed. Data are shown as the mean ± SD (n = 3).

Comment 5: Introduce the ACSS2 abbreviation already on Page 3, line 108, using it the first time.

Response 5:  We have corrected it accordingly.

Comment 6: Page 3, line 109: Delete the second dot. “in extracellular DNA..”

Response 6:  We have corrected it accordingly.

Comment 7: Page 4, line 118. Please provide a reference for the information “Neutrophils express monocarbonic acid transporter (MCT), an acetate transporter.”

Response 7: We added the references.

Comment 8: Point 2.5 and Figure 5: When was the histone acetylation measured of the control, which was not treated with sodium acetate? After 1h or 24h? That should be added.

Response 8: Control was not treated with sodium acetate, we added it in Figure 5.

Comment 9: Page 5, line 139. In the text you write, that “significant acetylation was observed at H3K9, H3K14, and H3K27” but in figure 6 you indicate significance only for H3K9, H3K14. Please clarify that. In the shown full Western blot of Figure 5 H3K27, the 1. experiment shows opposite results to the 2. experiment. In the 3. experiment the bands are looking very similar. I doubt an acetylation increase of H3K27 with CH3COONa. Please recheck this.

Response 9:  We thank the reviewer for these insightful comments. Since there was no statistically significant difference, I have rewritten the text “significant acetylation was observed at H3K9, H3K14, and H3K27” to “significant acetylation was observed at H3K9 and H3K14”

Comment 10: Figure 8: It would be easier to read and understand the treatments, if you label the treatment of A23187 an CH3COONa as you did it in figure 1b for the images. 

Response 10:  We have corrected it accordingly.

Specific comments to Supplemented Figures

Comment 11:I think Figure 3 should actually be named Figure 4. Is that correct? If so, please change it.

Response 11:  We have check it. We have corrected it accordingly.

Comment 12: Please check all the Figure numbers, there are some more mistakes.

Response 12:  We have check it.

Comment 13: Please indicate the size assignment of the protein marker.

Response 13: We have corrected it accordingly.

Round 2

Reviewer 1 Report

Comments and Suggestions for Authors

The manuscript improved the quality and the authors answered most of the questions I addressed. 

I would suggest revising the English, such as "Control was no treated sodium acetate". 

Author Response

We thank the reviewer for the careful review.

Comment 1: I would suggest revising the English, such as "Control was no treated sodium acetate". 

Response 1: As suggested, we have changed the sentence "Control was no treated sodium acetate." to “Control was not treated with sodium acetate.

Line 91,106,153,162,179, and 198